# Effect of hysterectomy on the risk of ovarian cancer: A South Korean national cohort study

**Jin-Sung Yuk**[1], **Sang-Hee Yoon**[2]*

**1** Department of Obstetrics and Gynecology, Sanggye Paik Hospital, School of Medicine, Inje University, Seoul, Republic of Korea, **2** Department of Obstetrics and Gynecology, Research Institute of Medical Science, Konkuk University School of Medicine, Seoul, Republic of Korea

* psimy81@gmail.com

## Abstract

### Background

Hysterectomy is a common gynecological surgery, but its long-term impact on ovarian cancer risk remains unclear, particularly in Asian populations.

### Objective

To evaluate the association between hysterectomy (with or without concomitant adnexal surgery) and the risk of ovarian cancer in South Korean women.

### Methods

We conducted a retrospective cohort study using the Korean National Health Insurance Service (NHIS) database (2002–2020). After 1:1 propensity score matching, 13,059 women who underwent hysterectomy for benign indications (aged 40–59) were compared with 13,059 women without hysterectomy. The primary outcome was incident ovarian cancer, defined by three or more medical visits with a C56.xx diagnosis code. Cox proportional hazards models were used to estimate hazard ratios (HRs) for ovarian cancer, adjusting for demographic and clinical confounders.

### Results

Over a median follow-up of 11.5 years, ovarian cancer incidence was 18 per 100,000 person-years in the hysterectomy group and 13 per 100,000 person-years in the non-hysterectomy group. Hysterectomy was associated with an imprecise estimate of ovarian cancer risk (HR 1.42, 95% CI 0.79–2.56), compatible with both a clinically meaningful decrease and increase in risk; therefore, the findings are inconclusive. There were no statistically significant differences between the two groups across various decades of life, including females below or above 50 years of age.

**Data availability statement:** The datasets used in this study (National Health Insurance Service–National Health Screening Cohort, NHIS-HEALS) are owned by the National Health Insurance Service (NHIS). Due to the NHIS's privacy policy, the raw data cannot be publicly shared. However, the data can be accessed through the National Health Insurance Sharing Service (NHISS) at https://nhiss.nhis.or.kr by researchers who apply for and receive institutional review board approval. After accessing https://nhiss.nhis.or.kr, the research number is NHIS-2022-1-309.

**Funding:** The author(s) received no specific funding for this work.

**Competing interests:** NO authors have competing interests.

## Conclusions

This study found no statistically significant association between hysterectomy and ovarian cancer risk, but the wide confidence intervals and limited number of events indicate that the findings remain inconclusive.

---

## 1. Introduction

Uterus excision or hysterectomy is a common surgical procedure used to treat gynecological morbidities such as abnormal bleeding, uterine fibroids, adenomyosis, and uterine prolapse in women typically impending or after menopause [1]. Although hysterectomy is a prevalent non-obstetric surgical procedure performed on women globally [2], the prevalence of hysterectomy varies between countries, with a gradual decrease observed in high-income countries and an increase observed in low and middle-income countries [1,3]. Many high-income countries report a decrease in prevalence of hysterectomy with rates ranging from 173/100,000 women in Denmark to 510/100,000 in the United States over time due to advances in alternative interventions [1]. Besides, there have been recent reports of an increase in hysterectomy among women from several regions of India [4]. According to the 2015−16 National Family Health Survey in India, almost 1 in 10 women had undergone hysterectomy by the age of 50, ranging up to 1 in 5 in the states of Andhra Pradesh and Telangana [5].

Hysterectomy affords prompt alleviation from the specific medical manifestation such as excessive bleeding and pelvic pain, but it has been ascertained to entail a multitude of enduring health ramifications [6,7]. A recent systematic review including 29 studies on hysterectomy with and without oophorectomy reported an association between hysterectomy and several chronic conditions, including increased risks of cardiovascular events, depression, metabolic disorders, and dementia [8]. Thus, the widespread use of hysterectomy represents an important issue for women's health over the life course.

Hysterectomy protects against cervical and uterine cancers, both of which affect the uterus.

Premenopausal hysterectomy, even with ovarian conservation, has been consistently associated with impaired ovarian function [9], likely due to disruption of the utero-ovarian blood supply and altered ovarian perfusion [10–12]. Reduced ovarian blood flow and lower levels of ovarian sex steroids have been observed after hysterectomy [13–15], and several cohort studies have reported almost a two-fold increase in the risk of ovarian failure and an earlier onset of menopause among women who undergo hysterectomy with ovarian preservation [10]. These changes suggest that hysterectomy may decrease cumulative lifetime ovulatory cycles, which in turn could reduce ovarian cancer risk according to the incessant ovulation hypothesis.

Conversely, several biological mechanisms support the possibility of a null or even increased risk after simple hysterectomy. Simple hysterectomy leaves the fallopian

tubes in situ, and accumulating pathologic and epidemiologic evidence indicates that many high-grade serous ovarian, fallopian tube, and primary peritoneal carcinomas arise from the distal fallopian tube epithelium rather than the ovarian surface itself [16,17]. Therefore, unlike salpingectomy or salpingo-oophorectomy, simple hysterectomy does not remove the presumed primary site of origin for these tumors. In addition, hysterectomy-induced earlier menopause may lead to prolonged use of estrogen-only menopausal hormone therapy (MHT) in women without a uterus, and estrogen-only MHT has been associated with an increased risk of ovarian cancer in observational studies [18]. These competing mechanisms suggest that the net effect of hysterectomy on ovarian cancer risk is biologically uncertain and could plausibly range from risk reduction to no effect or even risk increase.

Most epidemiologic studies on the association between hysterectomy and ovarian cancer risk have been conducted in Western populations, and their findings have been heterogeneous, with earlier studies generally suggesting a protective association and more recent analyses reporting weaker or even positive associations [19–21]. Data from Asian populations remain limited, and the overall direction and magnitude of the association between hysterectomy and ovarian cancer risk in these settings are still unclear. Given the high prevalence of hysterectomy and the emerging recognition of fallopian tube–origin high-grade serous carcinogenesis, clarifying the long-term ovarian cancer risk after simple hysterectomy in diverse racial and healthcare contexts is of considerable clinical importance. In this context, we aimed to evaluate the association between simple hysterectomy, with or without bilateral salpingo-oophorectomy, and the risk of ovarian cancer in a large, nationally representative cohort of Korean women. Rather than presuming a strictly protective effect, our study was designed to empirically assess whether hysterectomy is associated with a decreased, unchanged, or increased risk of ovarian cancer in this population.

## 2. Materials and methods

### 2.1. Study design and database

NHIS is a single-payer health care system that provides mandatory coverage to most residents of South Korea [22]. The NHIS database serves as a valuable source of health insurance information, encompassing crucial details such as the insured individuals' gender, age, medical coverage type, and the codes for their diagnoses, medications, and surgeries [22]. The NHIS also offers comprehensive health screening services to South Korean citizens at no cost, providing valuable measurement data and health history [22]. This population-based retrospective cohort study utilized data from the National Health Insurance and the National Health Checkup, both provided by the NHIS. The study covered the period from January 1, 2002, to December 31, 2020. The study participants were selected based on specific criteria verified by the 2020 edition of the Korea Health Insurance Medical Care Expenses and the International Classification of Diseases 10th revised edition (ICD-10). We accessed the data after receiving IRB approval for research purposes on December 27, 2021.

### 2.2. Participant selection

The subjects were divided into two groups: the hysterectomy group, which included women aged 40–59 years who had their uterus excised for benign reasons between January 1, 2003, and December 1, 2011, and the non-hysterectomy group, which comprised women aged 40–59 years who underwent a health checkup at the NHIS during the same period. A random sample of 25% of these women was selected to fit the capacity of the NHIS analytical server.

These participants were excluded from the study. These included women who underwent a health checkup or hysterectomy for benign reasons in 2002. Women with a medical diagnosis of cancer (any Cxx.xx) or benign ovarian disease (D27 (benign neoplasm of the ovary), N83.0 (follicular cyst of the ovary); N83.1 (corpus luteum cyst); N83.2 (other and unspecified ovarian cyst); N80.1 (endometriosis of the ovary); N80.2 (endometriosis of the fallopian tubes), N83.5 (torsion of the ovaries, cervix, and fallopian tubes)) within 365 days of study enrollment were excluded.

Propensity score matching with a 1:1 ratio was used to create the comparison groups. This matching method considered several variables such as age, alcohol consumption, physical exercise level, smoking status, hypertension, diabetes mellitus, dyslipidemia, socioeconomic status (SES), body mass index (BMI), parity, age at first menstruation, menopausal status, previous menopausal hormone therapy (MHT) use, residential area, Charlson comorbidity index (CCI) score, uterine fibroids, prior adnexal surgery, and endometriosis. The study followed up with the participants until December 31, 2020. Because this was a retrospective database study with a fixed study population determined by NHIS availability and eligibility criteria, we did not perform a pre-study sample size calculation.

## 2.3. Outcomes

Women with ovarian cancer were defined as those with three or more visits to a healthcare provider with a diagnosis code (C56.xx) for ovarian cancer in the primary or secondary diagnosis. Because the endpoint was defined using ICD-10 C56. xx claims codes, it captures malignant ovarian neoplasms coded as C56 and does not include borderline ovarian tumors typically coded as D39.1

## 2.4. Confounding variables

The following factors were examined in the study: age (in 5-year increments), BMI (using the criteria of the Asia-Pacific perspective) [23], SES (medical aid as medical insurance), smoking, alcohol consumption, and physical activity levels (self-reported measures), living area (urban or rural area), age at first menstruation (<13 years or ≥13 years), parity (0, 1, 2, or ≥3 births), menopausal status (determined through questionnaire responses), MHT use before inclusion (more than six months of tibolone, combined estrogen/progestogen, or estrogen alone before the study participation date), hysterectomy or concomitant adnexal surgery (recorded using surgery codes), diabetes mellitus (E10–E14), hypertension (I10–I15), uterine fibroids (D25), hyperlipidemia (E78), and endometriosis (N80) (determined by having visited a medical institution for the respective conditions two or more times before study participation) and CCI score (calculated from diagnosis codes from one year before the study participation date to the participation date) [24]. Concomitant adnexal surgery was defined using NHIS procedure codes indicating salpingectomy and/or oophorectomy performed at the time of hysterectomy. Because laterality is not reliably captured in the NHIS procedure codes available for this study, we could not distinguish unilateral from bilateral salpingectomy/oophorectomy.

## 2.5. Statistics

All statistical analyses were performed using R (version 3.5.1; The R Foundation for Statistical Computing, Vienna, Austria). A two-tailed significance level of p < 0.05 was set throughout. For baseline comparisons before matching, categorical variables were assessed using Pearson's chi-squared or Fisher's exact tests, and continuous variables were compared using Student's t-test and the Wilcoxon rank sum test as appropriate.

After propensity score matching, categorical variables were analyzed with the Cochran–Mantel–Haenszel test. Continuous variables were compared using the paired t-test and, given that the Anderson–Darling normality test indicated significant departures from normality for all variables (all p-values < 0.001), we relied primarily on the nonparametric Wilcoxon signed-rank test to ensure statistical validity. Covariate balance was evaluated using standardized mean differences.

To assess the effect of hysterectomy on ovarian cancer risk, stratified Cox regression analyses were performed, with the Schoenfeld residual test applied to check the proportional hazards assumption. For censoring, we used the earliest among death or last healthcare visit. List-wise deletion was used for missing data during matching. As a sensitivity analysis, we repeated Cox regression exclusively in women with a CCI score of 0. To address potential residual covariate imbalance after propensity score matching (particularly for age), we additionally fitted a multivariable Cox proportional hazards model in the matched cohort adjusting for all baseline covariates included in the propensity score model. All analyses were conducted between July 2022 and January 2023.

## 2.6. Ethics

This study was approved by the Institutional Review Board of Inje University Sanggye Paik Hospital (approval number: SGPAIK 2021-12-005). As these data are public and nonpersonally identifiable, the Institutional Review Board of Inje University Sanggye Paik Hospital IRB waived informed consent. The NHIS carefully anonymized the raw data and performed data analysis within its secure closed server, following its personal protection policy. To maintain confidentiality, unauthorized access to raw data was prevented by restricting data export solely to research outcome analysis. This measure ensured that only authorized individuals could access data and maintained confidentiality and integrity. Informed consent was not required for this study, in accordance with the Bioethics and Safety Act of South Korea. All methods were performed in accordance with the relevant guidelines and regulations.

## 3. Results

We identified 4,024,564 women who underwent hysterectomy or national health examinations between 2003 and 2011. Of these, 1,006,141 (one-fourth) were randomly selected. After propensity score matching, 13,059 women were allocated to the hysterectomy or non-hysterectomy group (Fig 1). The Anderson–Darling normality test applied to continuous covariates in the matched sample uniformly yielded p-values < 0.001, indicating non-normal distributions; therefore, we used the Wilcoxon signed-rank test for group comparisons. Covariate balance was confirmed, with all standardized mean differences (SMDs) below 0.2 (Table 1).

Most of the participants were in their late 40s (44.4%). They had a median age of 47 [45–50] years and a median follow-up of 11.5 [10–13.4] years (Table 1, S1 Table). The incidence of ovarian cancer was 18 cases per 100,000 person-years in the hysterectomy group and 13 cases per 100,000 person-years in the non-hysterectomy group (p = 0.312). These findings were further supported by a stratified log-rank test, which did not reveal any significant difference in the risk of ovarian cancer between the two groups (p = 0.2). A Kaplan–Meier plot of the incidence of ovarian cancer

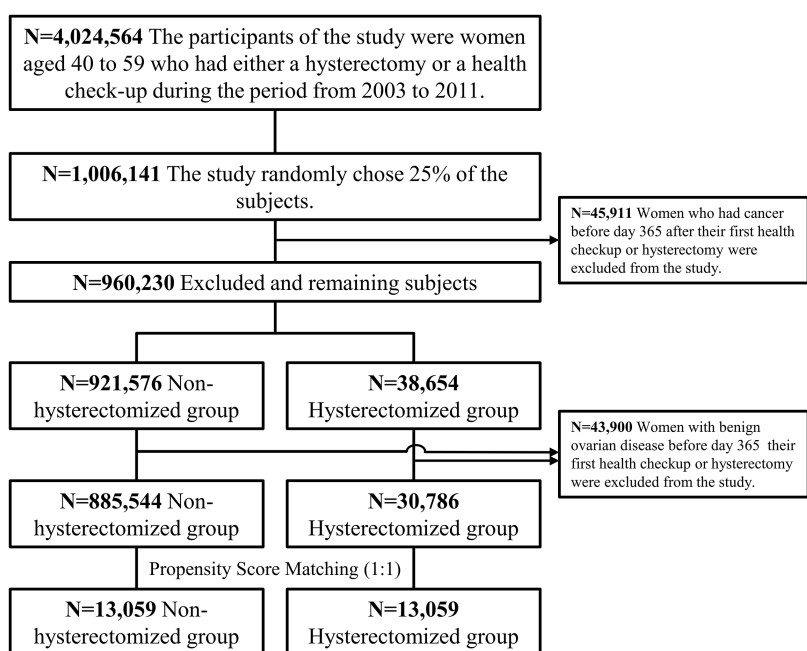

**Fig 1. Flowchart showing the process of selecting hysterectomy and non-hysterectomy groups using 2002-2020 Korean National Health Insurance data.**

**Table 1. Characteristics of the hysterectomy and non-hysterectomy groups among the subjects in this study (after propensity matching)).**

| | Non-Hysterectomy | Hysterectomy | Total | P-value | Standardized mean difference | Missing, % |
|---|---|---|---|---|---|---|
| Number of women | 13,059 | 13,059 | 26,118 | | | |
| Follow-up period (years) | 11.4 [10.1-13.5] | 11.5 [10-13.3] | 11.5 [10-13.4] | 0.89 | 0.010 | 0 |
| Median age (years) | 48 [44-50] | 47 [45-50] | 47 [45-50] | 0.397 | 0.058 | 0 |
| Age at inclusion (years) | | | | <0.001 | 0.174 | 0 |
| 40~44 | 3,350 (25.7) | 3,051 (23.4) | 6,401 (24.5) | | | |
| 45~49 | 5,274 (40.4) | 6,325 (48.4) | 11,599 (44.4) | | | |
| 50~55 | 3,544 (27.1) | 3,075 (23.5) | 6,619 (25.3) | | | |
| 55~60 | 891 (6.8) | 608 (4.7) | 1,499 (5.7) | | | |
| Year at inclusion | | | | <0.001 | 0.041 | 0 |
| 2003~2005 | 1,407 (10.8) | 1,505 (11.5) | 2,912 (11.1) | | | |
| 2006~2008 | 4,551 (34.8) | 4,313 (33) | 8,864 (33.9) | | | |
| 2009~2011 | 7,101 (54.4) | 7,241 (55.4) | 14,342 (54.9) | | | |
| Median BMI (kg/m$^2$) | 23.6 [21.7-25.8] | 23.6 [21.8-25.8] | 23.6 [21.8-25.8] | 0.685 | 0.001 | 0 |
| BMI (kg/m2) | | | | <0.001 | 0.041 | 0 |
| <18.5 | 238 (1.8) | 194 (1.5) | 432 (1.7) | | | |
| 18.5-22.9 | 5,102 (39.1) | 5,220 (40) | 10,322 (39.5) | | | |
| 23-24.9 | 3,276 (25.1) | 3,368 (25.8) | 6,644 (25.4) | | | |
| 25-29.9 | 3,866 (29.6) | 3,747 (28.7) | 7,613 (29.1) | | | |
| ≥30 | 577 (4.4) | 530 (4.1) | 1,107 (4.2) | | | |
| Low SES | 47 (0.4) | 64 (0.5) | 111 (0.4) | 0.09 | 0.020 | 0 |
| Rural area | 9,317 (71.3) | 9,372 (71.8) | 18,689 (71.6) | 0.275 | 0.009 | 0 |
| CCI | | | | 0.128 | 0.019 | 0 |
| 0 | 10,585 (81.1) | 10,643 (81.5) | 21,228 (81.3) | | | |
| 1 | 2,030 (15.5) | 1,949 (14.9) | 3,979 (15.2) | | | |
| ≥2 | 444 (3.4) | 467 (3.6) | 911 (3.5) | | | |
| Parity | | | | 0.028 | 0.027 | 0 |
| 0 | 1,890 (14.5) | 1,919 (14.7) | 3,809 (14.6) | | | |
| 1 | 1,687 (12.9) | 1,585 (12.1) | 3,272 (12.5) | | | |
| 2 | 8,738 (66.9) | 8,846 (67.7) | 17,584 (67.3) | | | |
| ≥3 | 744 (5.7) | 709 (5.4) | 1,453 (5.6) | | | |
| Age over 13 at menarche (years) | 10,330 (79.1) | 10,267 (78.6) | 20,597 (78.9) | 0.169 | 0.012 | 0 |
| Menopause before inclusion | 1,992 (15.3) | 1,869 (14.3) | 3,861 (14.8) | <0.001 | 0.027 | 0 |
| Smoking | | | | 0.023 | 0.030 | 0 |
| Never | 12,216 (93.5) | 12,307 (94.2) | 24,523 (93.9) | | | |
| Past | 241 (1.8) | 205 (1.6) | 446 (1.7) | | | |
| Current | 602 (4.6) | 547 (4.2) | 1,149 (4.4) | | | |
| Alcohol (per week) | | | | <0.001 | 0.063 | 0 |
| None | 8,975 (68.7) | 9,196 (70.4) | 18,171 (69.6) | | | |
| ~2/week | 3,843 (29.4) | 3,569 (27.3) | 7,412 (28.4) | | | |
| 3~6/week | 154 (1.2) | 221 (1.7) | 375 (1.4) | | | |
| Daily | 87 (0.7) | 73 (0.6) | 160 (0.6) | | | |
| Physical exercise (per week) | | | | 0.304 | 0.019 | 0 |
| None | 8,357 (64) | 8,419 (64.5) | 16,776 (64.2) | | | |
| 1~2 | 2,576 (19.7) | 2,499 (19.1) | 5,075 (19.4) | | | |
| 3~4 | 1,268 (9.7) | 1,288 (9.9) | 2,556 (9.8) | | | |
| 5~6 | 416 (3.2) | 394 (3) | 810 (3.1) | | | |

*(Continued)*

**Table 1.** (Continued)

|  | Non-Hysterectomy | Hysterectomy | Total | P-value | Standardized mean difference | Missing, % |
|---|---|---|---|---|---|---|
| Daily | 442 (3.4) | 459 (3.5) | 901 (3.4) |  |  |  |
| DM before inclusion | 1,352 (10.4) | 1,291 (9.9) | 2,643 (10.1) | 0.212 | 0.015 | 0 |
| Hypertension before inclusion | 2,497 (19.1) | 2,500 (19.1) | 4,997 (19.1) | 0.974 | 0.001 | 0 |
| Dyslipidemia before inclusion | 2,093 (16) | 2,045 (15.7) | 4,138 (15.8) | 0.414 | 0.010 | 0 |
| MHT before inclusion | 300 (2.3) | 168 (1.3) | 468 (1.8) | <0.001 | 0.076 | 0 |
| Adnexal surgery before inclusion | 39 (0.3) | 11 (0.1) | 50 (0.2) | <0.001 | 0.049 | 0 |
| Uterine fibroids before inclusion | 9,776 (74.9) | 9,705 (74.3) | 19,481 (74.6) | <0.001 | 0.012 | 0 |
| Endometriosis before inclusion | 2,139 (16.4) | 2,205 (16.9) | 4,344 (16.6) | <0.001 | 0.014 | 0 |

DM, diabetes mellitus; CCI, Charlson comorbidity index; MHT, menopausal hormone therapy; SES, socioeconomic status.

The data is presented as either a number and percentage, or as a median with the interquartile range.

in the two groups is shown in Fig 2. The near-overlapping curves should be interpreted as reflecting limited event information and imprecision rather than evidence of no effect.

In our stratified Cox proportional analysis assessing the risk of ovarian cancer, the hysterectomy and non-hysterectomy groups exhibited comparable risks of developing ovarian cancer, with a HR of 1.42 and a 95% CI of 0.79 to 2.56 (Table 2). The wide confidence interval indicates substantial uncertainty, and the data remain compatible with both risk reduction and risk increase. Subgroup analyses were exploratory and were not used for inference because of sparse events in some strata (e.g., concomitant adnexal surgery subgroup). Therefore, only the primary matched analysis is emphasized in the main Results. The Cox proportional hazards model was used to analyze the relationship between hysterectomy and ovarian cancer incidence based on age strata. Notably, there were no statistically significant differences between the two groups across various decades of life, including women below or above 50 years of age. For instance, the hazard ratio for women in their 40s stood at 1.17 (95% CI 0.54 to 2.52) and for participants entering their fifth decade it was 1.14 (CI 0.41 to 3.15) respectively. The entire dataset is presented in S2 Table.

Regarding Table 1, which contains information on various patient characteristics, S3 Table presents additional specifics on ovarian cancer incidence rates. To ensure thoroughness, we also conducted a sensitivity analysis focusing specifically on patients who had zero CCI entries, that is, those likely harboring fewer comorbid illnesses than others suffering from chronic conditions in the general population. Because age remained the largest residual imbalance after matching (SMD 0.174), we performed a multivariable Cox regression in the matched cohort additionally adjusting for all baseline covariates (including age). The adjusted hazard ratios remained not statistically significant: hysterectomy (with/without concomitant adnexal surgery) HR 1.22 (95% CI 0.56–2.68; P = 0.621), hysterectomy without concomitant adnexal surgery HR 1.16 (0.49–2.73; P = 0.732), and hysterectomy with concomitant adnexal surgery HR 1.66 (0.17–15.79; P = 0.659). Our reanalysis revealed consistent findings whereby hysterectomy's presence or absence does not appreciably modify ovarian cancer incidence rate among these low complexity individuals either (HR 1.53, 95% CI 0.80–2.94). Moreover, the primary conclusion of this study was true regardless of whether the patients were severely unwell due to other health issues.

## 4. Discussion

Using national population-based cohort data, this study did not find a statistically significant association between hysterectomy and ovarian cancer risk among middle-aged and older women in Korea (HR 1.42, 95% CI 0.79–2.56), but the wide confidence intervals indicate considerable uncertainty regarding the true magnitude and even the direction of the association. Although the association did not reach statistical significance, the point estimate (HR 1.42) suggests a possible increase in ovarian cancer risk after hysterectomy. If this estimate reflected the true effect, it would be clinically important.

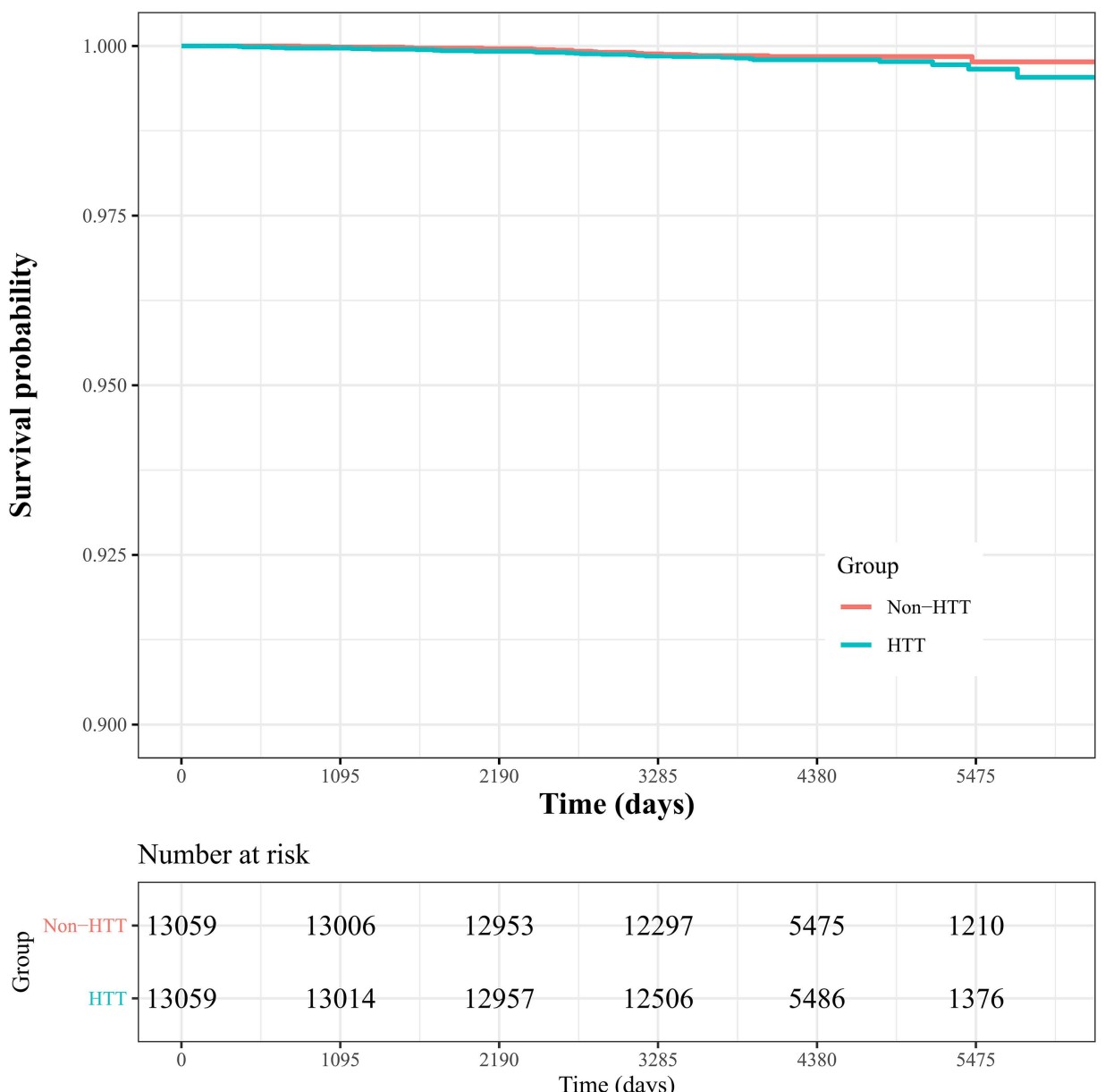

**Fig 2. Kaplan-Meier plots of ovarian cancer incidence in the hysterectomy and non-hysterectomy groups (stratified log-rank test p-value = 0.2).** HTT, hysterectomy.

**Table 2. Risk of ovarian cancer in women who had a hysterectomy using Cox proportional hazards analysis.**

|  | Ovarian cancer cases per 100,000 person-years | HR (95% CI) | |
|---|---|---|---|
|  |  | HR (95% CI) [a] | P-value |
| Non-hysterectomy | 20/154,403 (13) | 1 | |
| Hysterectomy | 28/154,109 (18) | 1.42 (0.79-2.56) | 0.241 |

CI, confidence interval; HR, hazard ratio.

[a]Because propensity score matching was performed, hazard ratios were not adjusted for additional confounders.

However, the wide 95% confidence interval (0.79–2.56) indicates that our data are also compatible with a modest risk reduction and with a substantially larger risk increase. Given the limited number of ovarian cancer events in our cohort, our study cannot reliably distinguish among these possibilities and therefore cannot exclude a clinically meaningful increase in risk. There were no statistically significant differences between the two groups across various decades of life, including females below or above 50 years of age.

Comprehensive studies have been conducted on the association between ovarian cancer and various gynecologic surgeries, including hysterectomy, salpingectomy, tubal ligation, oophorectomy, and salpingo-oophorectomy [19,25]. Up to now, the surgical removal of fallopian tubes and ovaries has been found to be inversely associated with ovarian cancer. Several studies have also suggested that high-grade serous ovarian, fallopian tube, and primary peritoneal carcinomas are comprised of cells that resemble the fallopian tube epithelium [16]. Based on these findings, the tubal hypothesis, which suggests that high-grade serous cancers could originate in the fallopian tube, was introduced in 2010 [17]. The potential paradigm shift in the comprehension of ovarian cancer is reinforced by epidemiological data indicating that the occlusion or removal of the fallopian tubes protects against subsequent ovarian cancer development [26,27]. Hysterectomy as well as salpingectomy may alter ovarian cancer risk by protecting the ovary from ascending carcinogens. In addition, the impact of premenopausal hysterectomy on ovarian function is associated with disrupted utero-ovarian artery blood flow and reduced sex steroids, potentially leading to premature ovarian failure [9]. Hysterectomy alone increases ovarian failure risk by nearly 2-fold and causes earlier menopause [10]. This diminished ovarian function could potentially reduce the risk of ovarian cancer by decreasing the frequency of ovulatory cycles. Repeated exposure of the ovaries and tubes to the physical and hormonal effects of ovulation (incessant ovulation) is considered one of the most frequently mentioned theories explaining the development of ovarian cancer [28].

To our knowledge, there are three meta-analyses evaluating the association between hysterectomy and ovarian cancer risk [19–21]. The former meta-analysis included 24 observational studies conducted in 2012, and the results showed that hysterectomy is associated with a decrease in the risk of ovarian cancer [19]. The majority of papers included in this meta-analysis were case-control studies, and subgroup analysis was performed on both simple hysterectomy and hysterectomy with unilateral oophorectomy. The summary relative risk (RR) for women with vs. without hysterectomy was 0.74 (95% CI: 0.65–0.84). Simple hysterectomy and hysterectomy with unilateral oophorectomy were associated with a similar decrease in risk (summary relative risk: 0.62, 95% CI: 0.49–0.79 and 0.60, 95% CI:0.47–0.78, respectively). The latter meta-analysis including 20 observational study was conducted in 2013 [20]. The overall relative risk estimate was 0.81 (95% CI 0.72–0.92) suggesting hysterectomy decreases the risk of ovarian cancer. However, significant heterogeneity was observed (I2 = 74%). The authors assessed the effects of possible sources of heterogeneity, including study design, country, response rate in controls, prevalence of hysterectomy, and median year of diagnosis (pre- versus post-2000) using subgroup analyses and multivariate meta-regression models. The largest difference was observed for median year of diagnosis with a pooled relative risk of 0.70 (95% CI: 0.65–0.76) for studies with a median year of diagnosis pre-2000, versus 1.18 (95% CI: 1.06–1.31) post-2000. The results showed a temporal shift in the association between hysterectomy and the risk of ovarian cancer. Specifically, studies in which most women were diagnosed before 2000 found that hysterectomy was associated with a lower risk of ovarian cancer, whereas studies in which the majority of women were diagnosed after 2000 showed a positive association. The most recent meta-analysis was conducted to investigate the long-term outcomes, such as cardiovascular events, metabolic disorders, cancer, depression, and dementia, of hysterectomy with bilateral salpingo-oophorectomy (BSO) [21]. To assess the risk of ovarian cancer among these various outcomes in the meta-analysis, five studies published after 2009 were included [16,25,29–31]. Three studies evaluated the association of hysterectomy with BSO and ovarian cancer risk, including women who had undergone hysterectomy as the reference group [16,29,31]. This approach allowed them to examine the risks associated with elective BSO but limited their ability to separately evaluate the relative risks associated with simple hysterectomy. Conversely, the other two studies investigated the associations of both simple hysterectomy and hysterectomy with BSO in connection to ovarian cancer

risk, in comparison to no surgery [25,30]. Associations with simple hysterectomy are plausible due to the potential of the procedure to impair ovarian circulation [32,33]. The reduction in ovarian cancer risk was observed in this meta-analysis of women with previous hysterectomy with BSO on benign indication (HR: 0.11, 95% CI: 0.09–0.15). In detail, hysterectomy with BSO compared to hysterectomy alone or no surgery was associated with substantial reductions in ovarian cancer risk (HR: 0.14, 95% CI: 0.09–0.20 and 0.09, 95% CI: 0.06–0.14, respectively).

Contrary to most previous studies, our analysis of ovarian cancer risk related to hysterectomy, using non-surgical controls as a reference group, did not show a significant reduction in ovarian cancer. Several plausible explanations may account for this finding. First, our findings align with the 'temporal shift' observed in a 2013 meta-analysis, which reported protective effects (pooled RR 0.70) in pre-2000 studies but increased risk (RR 1.18) in post-2000 studies. This pattern may reflect improved diagnostic specificity in distinguishing ovarian cancer. Second, this discrepancy reflects indication bias: women requiring hysterectomy often have underlying conditions such as endometriosis or adenomyosis, both of which are recognized risk factors for ovarian cancer [34], but these factors may not be fully captured by claims-based propensity score matching. Third, our study participants were women aged 40–59 years with a median age of 47 [45–50] years at hysterectomy and a median follow-up of 11.5 [10–13.4] years. Notably, prior analyses have shown that hysterectomies performed closer to the date of diagnosis show no association with ovarian cancer risk, whereas those performed ≥20 years prior are associated with a 30% risk reduction (RR 0.71, 95% CI 0.55–0.90) [20]. Thus, our relatively short post-hysterectomy follow-up may have limited our ability to detect protective effects. Finally, the potential explanation for our findings pertains to the use of menopausal hormone therapy (MHT) in women experiencing menopausal symptoms following hysterectomy. Hysterectomy could potentially induce premature menopause, resulting in a higher probability of estrogen-only hormone replacement therapy being prescribed for extended durations among women undergoing hysterectomy. Furthermore, women who have undergone hysterectomy are predominantly prescribed estrogen without progestin, and estrogen-only MHT has been associated with an increased risk of ovarian cancer [18]. The adoption of postoperative estrogen-only MHT might have counterbalanced the reduction in ovarian cancer risk associated with hysterectomy.

This study has several notable strengths. First, it utilizes a large, nationally representative cohort drawn from the comprehensive Korean National Health Insurance Service database, ensuring broad generalizability to South Korean women. The dataset's high quality and longitudinal design, featuring a substantial sample size and extended follow-up, enhance statistical power and enable robust outcome assessment. Confounding factors were rigorously controlled through strict inclusion and exclusion criteria and detailed propensity score matching across a wide range of demographic and clinical variables, with balance confirmed by standardized mean differences (all SMDs < 0.2). Finally, most studies on this topic have been conducted in Western countries, including the USA, Canada, Denmark, Sweden, the Netherlands, and Finland. In contrast, our study pioneered research in Asian populations.

Nonetheless, some inherent limitations merit acknowledgment. The most important limitation of this study is the limited number of ovarian cancer events, which resulted in wide confidence intervals and low statistical power. Using Schoenfeld's method for Cox models (two-sided $\alpha = 0.05$), ~247 events would be required for 80% power to detect HR = 0.7 and ~456 events for HR = 1.3; with only 48 events, power is ~23% and ~15%, respectively, and the study is only powered for very large effects (approximately HR < 0.45 or >2.24). In particular, the study was severely underpowered to detect anything but a very large effect size, which fundamentally limits the conclusiveness of our findings and means that modest increases or decreases in risk cannot be ruled out. Exploratory subgroup estimates were unstable due to sparse outcome events in some strata (e.g., concomitant adnexal surgery subgroup), and therefore were not emphasized for inference. In addition, outcome misclassification is possible because ovarian cancer was ascertained from claims data and we did not link our analytic dataset directly to the Korean Central Cancer Registry (KCCR). However, a Korean validation study using a KCCR–NHIS linked database reported that operational definitions based on NHIS claims showed high sensitivity (>90%) for major cancers when using either primary diagnosis codes or RID registration claims, and that >80% of patients had a < 31-day difference between the registry diagnosis date and the RID-based claims definition [35]. That study also

suggested that the accuracy of RID-based definitions improved after the mid-2000s as the RID program became established, which should be considered when interpreting claims-based case definitions across earlier calendar years. We lacked information on individual genetic risk (e.g., BRCA mutations) and family history of ovarian cancer, restricting exploration of high-risk populations. Although we applied propensity score matching using socioeconomic and lifestyle variables from the same screening program, residual confounding related to health-seeking behavior may still remain. As a retrospective cohort analysis, the potential for unmeasured confounding remains, despite comprehensive covariate adjustment. Lastly, our cohort consisted exclusively of Korean women, which – while relevant for Asian populations – may limit direct extrapolation of findings to other ethnic groups.

Future prospective studies with ≥20-year follow-up are needed to assess whether protective effects of hysterectomy on ovarian cancer risk emerge only after longer latency periods, as suggested by prior analyses showing risk reduction primarily among women ≥20 years post-hysterectomy [20].

In conclusion, Hysterectomy was associated with an imprecise, statistically non-significant estimate of ovarian cancer risk; due to limited events and wide confidence intervals, the findings are inconclusive. Further research is needed before clinical recommendations regarding hysterectomy for ovarian cancer risk reduction can be confidently made.

## Supporting information

**S1 Table. Characteristics of the hysterectomy and non-hysterectomy groups among the subjects in this study (before propensity matching).**
(DOCX)

**S2 Table. Exploratory risk of ovarian cancer by age in women who had a hysterectomy using Cox proportional hazards analysis.**
(DOCX)

**S3 Table. Analysis of ovarian cancer incidence per 100,000 person-years in women with and without hysterectomy using Korean National Health Insurance data from 2002-2020.**
(DOCX)

## Author contributions

**Conceptualization:** Jin-Sung Yuk, Sang-Hee Yoon.

**Data curation:** Jin-Sung Yuk.

**Methodology:** Jin-Sung Yuk.

**Resources:** Jin-Sung Yuk.

**Software:** Jin-Sung Yuk.

**Supervision:** Jin-Sung Yuk.

**Writing – original draft:** Sang-Hee Yoon.

**Writing – review & editing:** Sang-Hee Yoon.

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
