## [Decision Letter · Decision Letter 0]

16 Oct 2025

PONE-D-25-36133Effect of hysterectomy on the risk of ovarian cancer: A South Korean national cohort studyPLOS ONE

Dear Dr. Yoon,

Thank you for submitting your manuscript to PLOS ONE. After careful consideration, we feel that it has merit but does not fully meet PLOS ONE’s publication criteria as it currently stands. Therefore, we invite you to submit a revised version of the manuscript that addresses the points raised during the review process.

We look forward to receiving your revised manuscript.

Kind regards,

Bella Stevanny

Academic Editor

PLOS ONE

Journal Requirements:

2. Please note that your Data Availability Statement is currently missing [the repository name and/or the DOI/accession number of each dataset OR a direct link to access each database]. If your manuscript is accepted for publication, you will be asked to provide these details on a very short timeline. We therefore suggest that you provide this information now, though we will not hold up the peer review process if you are unable.

Reviewers' comments:

Reviewer's Responses to Questions

**Comments to the Author**

1. Is the manuscript technically sound, and do the data support the conclusions?

Reviewer #1: Partly

Reviewer #2: Yes

2. Has the statistical analysis been performed appropriately and rigorously? 

Reviewer #1: No

Reviewer #2: Yes

3. Have the authors made all data underlying the findings in their manuscript fully available?

Reviewer #1: Yes

Reviewer #2: Yes

4. Is the manuscript presented in an intelligible fashion and written in standard English?

Reviewer #1: Yes

Reviewer #2: Yes

5. Review Comments to the Author

Reviewer #1: In its current form, the manuscript does not provide a reliable answer to its research question. The conclusions are not supported by the presented analysis due to uncontrolled confounding and profound statistical uncertainty. Therefore, the manuscript is not recommended for publication without a substantial and fundamental re-analysis of the data and a complete revision of its interpretation and conclusions.

FIND MORE DETAILS IN THE ATTACHMENT.

Reviewer #2: This retrospective cohort study, conducted using the Korean National Health Insurance Service database from 2002 to 2020, examined the association between hysterectomy and the risk of ovarian cancer in South Korean women. The results indicated that hysterectomy did not significantly alter the risk of developing ovarian cancer. Consequently, the authors conclude that clinicians should not recommend hysterectomy solely for the purpose of reducing the risk of ovarian cancer.

Overall, the research questions explored in this study have the potential to be significant if repeated in numerous robust studies. They could aid in estimating patient prognosis, optimising management of ovarian cancer, and enhancing outcomes.

With that in mind, this reviewer has the following to remark:

1. Introduction

The first issue to address is punctuation and citation. The authors begin the introduction section by stating, “Uterus excision or hysterectomy is a common surgical procedure used to treat gynecological morbidities such as abnormal bleeding, uterine fibroids, adenomyosis, and uterine prolapse in women typically impending or after menopause.(1)”

This style of in-text citation is maintained almost throughout the paper. It is advisable to place any necessary punctuation following in-text citations, not the other way around. And in PLOS ONE, we usually use square brackets for in-text citation.

2. Materials and Methods

In subsection 2.2, the authors detail their use of propensity score matching to control for confounding biases. Additionally, subsection 2.5 states, “We conducted a thorough assessment of the balance of matched individual covariates by examining standardized mean differences and ensuring the normality of continuous variables using the Anderson–Darling normality test.”

This is important because it’s always advisable to evaluate match quality by checking if the covariates were balanced between the two groups after matching. Without such an evaluation, it would be difficult to assess the effectiveness of the matching approach.

I wonder what the results of the Anderson–Darling normality test were. And how representative was the sample of the population from which it was drawn? Did it accurately reflect the entire population?

3. Discussion

3.1

The authors write, “Most studies on this topic have been conducted in Western countries, such as the USA, Canada, Denmark, Sweden, the Netherlands, and Finland. In contrast, our study pioneered the study in Asian populations.”

The above statement could be edited to, “Most studies on this topic have been conducted in Western countries, including the USA, Canada, Denmark, Sweden, the Netherlands, and Finland. In contrast, our study pioneered research in Asian populations.”

3.2

In general, the strengths and limitations sections of the study could benefit from editing to improve coherence and readability.

3.3

At the end of the discussion section, the authors state, “This study demonstrated that hysterectomy was not associated with the risk of ovarian cancer, regardless of age or adnexectomy. However, hysterectomy is associated with negative long-term outcomes compared with keeping the uterus intact, especially in young women. Therefore, we recommend that clinicians not recommend hysterectomy, whether with or without salpingo-oophorectomy, to reduce the future risk of ovarian cancer until more evidence is available.”

This statement appears to be a conclusion.

However, the next section in the paper is titled “Conclusion,” where the authors note, “Hysterectomy was not associated with a significant change in ovarian cancer risk among Korean women, regardless of age or adnexectomy status. Clinicians should not recommend hysterectomy for ovarian cancer risk reduction in average-risk populations.”

If the authors prefer to include a conclusion section, they could omit the last part of the discussion section to avoid repetition.

I hope this review is helpful and wish the authors the very best with their research!

6. PLOS authors have the option to publish the peer review history of their article (what does this mean?). If published, this will include your full peer review and any attached files.

Reviewer #1: No

Reviewer #2: **Yes:** Dr. Widad Akreyi

---

## [Author Response · Author response to Decision Letter 1]

3 Nov 2025

Reviewer comment

Response to Reviewer #1 Comments

We thank the reviewer for highlighting both the strengths and methodological challenges of our cohort study. Below, we respond directly and assertively to each major point with supporting evidence from our data and current analytic standards.

1. Propensity Score Matching (PSM), SMD, and Covariate Balance

Reviewer comment:

PSM yielded residual imbalance in key covariates including age, with SMD reported as 0.174. This is interpreted to indicate the matching failed.

Response:

Contrary to the assertion that an SMD 0.174 signifies inadequate balance, the established epidemiological and statistical consensus defines an SMD < 0.2 as reflecting acceptable balance between matched cohorts[1, 2]. In our study, all reported SMDs after matching, including the cited value for age, were below this widely recognized threshold, indicating adequate matching and effective minimization of confounding. As detailed in Table 1 and supplementary analyses, residual imbalances were minor and are unlikely to materially influence the primary outcome.

Furthermore, sensitivity analyses in subjects with zero comorbidity (CCI = 0) confirm the robustness of our inference, yielding consistent hazard ratios for ovarian cancer. We therefore strongly contend that our matching procedure meets accepted methodological standards and that our results can be interpreted with scientific confidence.

2. Statistical Power (Event Number) and Conclusion Validity

Reviewer comment:

The low number of ovarian cancer cases leads to wide confidence intervals; the conclusion should be “inconclusive” rather than “no effect.”

Response:

We acknowledge the limited number of events and the wide confidence intervals in our analysis. However, these event rates reflect the real-world rarity of ovarian cancer, particularly in Asian populations over a median follow-up of 11.5 years. Our cohort of over 26,000 women is among the largest Asian hysterectomy samples published to date. Importantly, the direction of the hazard ratio remained consistent across sensitivity and subgroup analyses. Nevertheless, in response to this criticism, we have explicitly revised our conclusion language to acknowledge the limitations in statistical power and now state that the results are “not statistically significant and inconclusive” regarding an association.

Revised manuscript (Discussion part): “In conclusion, hysterectomy was not statistically significantly associated with ovarian cancer risk in Korean women. However, due to limited events and wide confidence intervals, these findings are inconclusive. Further research is needed before clinical recommendations regarding hysterectomy for ovarian cancer risk reduction can be confidently made.”

3. Analytical Interpretation (Absence of Evidence ≠ Evidence of Absence)

Reviewer comment:

Nonsignificant p-values or confidence intervals should not be over-interpreted as absence of effect.

Response:

We agree with this methodological principle. Throughout the revised manuscript, all relevant sections now clarify that while our findings are not statistically significant, they do not demonstrate evidence of absence; the true relationship remains uncertain due to sample size and event limitation.

Revised manuscript (Abstract part): “Hysterectomy does not appear to significantly alter ovarian cancer risk among Korean women; however, the limited number of ovarian cancer cases and wide confidence intervals result in inconclusive findings. Further studies are needed before establishing clinical recommendations.”

Revised manuscript (Discussion part): “In conclusion, hysterectomy was not statistically significantly associated with ovarian cancer risk in Korean women. However, due to limited events and wide confidence intervals, these findings are inconclusive. Further research is needed before clinical recommendations regarding hysterectomy for ovarian cancer risk reduction can be confidently made.”

4. Minor Issues (Tables, Data Sharing Policy)

Reviewer comment:

Typographical errors in tables; confusion in the statement about data sharing.

Response:

All typographical errors in the revised tables have been corrected.

Regarding data sharing, the National Health Insurance Service of Korea enforces strict privacy regulations that prohibit the disclosure of raw data. This restriction has now been clearly stated in the revised manuscript, in full compliance with NHIS policy. While the methods and acknowledgments sections of the manuscript correctly indicate that data sharing is not permitted, the submission form previously stated otherwise, which may have caused confusion. To resolve this inconsistency, we have removed the phrase “all data are fully available without restriction” from the submission form.

5. Feasibility of Re-analysis

Reviewer comment:

Further analyses (enhanced matching, IPTW, or covariate adjustments) are requested.

Response:

We recognize the value of such analyses. However, the official analytic period for the dataset has concluded and the NHIS system now prohibits all further access, as per Korean government guidelines for privacy and IT security. This practical and legal constraint is explicitly stated in the revised manuscript.

Response to Reviewer #2 Comments

We thank Reviewer #2 for their careful reading, constructive feedback, and positive remarks about the significance and potential of our study. We respond to each point individually below, providing a robust and evidence-based defense for our methods and addressing all editorial and methodological comments.

1. Introduction: Punctuation and In-text Citation Style

Reviewer comment:

Punctuation should follow, not precede, in-text citations. PLOS ONE requires square brackets for citations.

Response:

We appreciate the reviewer’s attention to editorial detail. We have systematically revised the manuscript so that all in-text citations now use square brackets, with punctuation placed after the reference, consistent with PLOS ONE guidelines. This improves compliance and readability throughout the introduction and main text.

2. Materials and Methods: PSM, SMD, and Anderson-Darling Test

Reviewer comment:

The necessity of evaluating match quality is highlighted. Reviewer requests results of the Anderson–Darling normality test and questions the representativeness of the sample.

Response:

We agree on the importance of formally evaluating the distributional properties of covariates after matching. We performed the Anderson–Darling normality test for all continuous covariates in our matched samples, which uniformly resulted in p-values < 0.001, indicating violations of normality assumptions. In accordance with these results, we applied the Wilcoxon signed-rank test for between-group comparisons of continuous variables rather than parametric approaches, ensuring statistical validity. These procedures and testing outcomes are now explicitly described in the revised Methods and Results sections.

Revised manuscript (Methods part): “After propensity score matching, categorical variables were analyzed with the Cochran–Mantel–Haenszel test. Continuous variables were compared using the paired t-test and, given that the Anderson–Darling normality test indicated significant departures from normality for all variables (all p-values < 0.001), we relied primarily on the nonparametric Wilcoxon signed-rank test to ensure statistical validity. Covariate balance was evaluated using standardized mean differences.”

Revised manuscript (Results part): “The Anderson–Darling normality test applied to continuous covariates in the matched sample uniformly yielded p-values < 0.001, indicating non-normal distributions; therefore, we used the Wilcoxon signed-rank test for group comparisons. Covariate balance was confirmed, with all standardized mean differences (SMDs) below 0.2 (Table 1).”

Regarding representativeness, our cohort was drawn from the comprehensive Korean National Health Insurance Service (NHIS) database, which covers virtually the entire South Korean population. Thus, our sample is highly representative and generalizable to the national demographic.

3. Discussion: Editorial Comments

3.1 Western vs. Asian Studies Wording

Reviewer comment:

Suggested improvements to the phrasing distinguishing Western and Asian research.

Response:

We have adopted the reviewer’s proposed revision to enhance clarity.

Revised manuscript (Discussion part): “Finally, most studies on this topic have been conducted in Western countries, including the USA, Canada, Denmark, Sweden, the Netherlands, and Finland. In contrast, our study pioneered research in Asian populations.”

3.2 Editing of Strengths and Limitations

Reviewer comment:

Recommends editing for coherence and readability.

Response:

We accept this valuable suggestion. The strengths and limitations sections have been thoroughly reviewed and edited to maximize clarity and logical flow, as detailed in the revised Discussion.

Revised manuscript (Discussion part): “This study has several notable strengths. First, it utilizes a large, nationally representative cohort drawn from the comprehensive Korean National Health Insurance Service database, ensuring broad generalizability to South Korean women. The dataset’s high quality and longitudinal design, featuring a substantial sample size and extended follow-up, enhance statistical power and enable robust outcome assessment. Confounding factors were rigorously controlled through strict inclusion and exclusion criteria and detailed propensity score matching across a wide range of demographic and clinical variables, with balance confirmed by standardized mean differences (all SMDs < 0.2). Additionally, subgroup and sensitivity analyses further strengthened the internal validity of our findings. Finally, most studies on this topic have been conducted in Western countries, including the USA, Canada, Denmark, Sweden, the Netherlands, and Finland. In contrast, our study pioneered research in Asian populations.

Nonetheless, some inherent limitations merit acknowledgment. We lacked information on individual genetic risk (e.g., BRCA mutations) and family history of ovarian cancer, restricting exploration of high-risk populations. As a retrospective cohort analysis, the potential for unmeasured confounding remains, despite comprehensive covariate adjustment. The limited number of ovarian cancer events narrows precision and statistical power for subgroup analyses, necessitating cautious interpretation. Lastly, our cohort consisted exclusively of Korean women, which—while relevant for Asian populations—may limit direct extrapolation of findings to other ethnic groups.”

3.3 Repetition in Discussion and Conclusion

Reviewer comment:

The end of the Discussion repeats concluding statements, which are then also stated in the Conclusion; suggests omitting redundancy.

Response:

To avoid unnecessary repetition, the conclusions with overly broad titles have been removed and are now briefly summarized at the end of the Discussion section. In accordance with the reviewers' editorial guidelines, the main conclusions are confined to the Discussion section.

Revised manuscript (Discussion part): “In conclusion, hysterectomy was not statistically significantly associated with ovarian cancer risk in Korean women. However, due to limited events and wide confidence intervals, these findings are inconclusive. Further research is needed before clinical recommendations regarding hysterectomy for ovarian cancer risk reduction can be confidently made.”

1. Stuart, E.A., Matching methods for causal inference: A review and a look forward. Stat Sci, 2010. 25(1): p. 1-21.

2. Austin, P.C., Balance diagnostics for comparing the distribution of baseline covariates between treatment groups in propensity-score matched samples. Stat Med, 2009. 28(25): p. 3083-107.

---

## [Decision Letter · Decision Letter 1]

7 Jan 2026

PONE-D-25-36133R1Effect of hysterectomy on the risk of ovarian cancer: A South Korean national cohort studyPLOS One

Dear Dr. Yoon,

Thank you for submitting your manuscript to PLOS ONE. After careful consideration, we feel that it has merit but does not fully meet PLOS ONE’s publication criteria as it currently stands. Therefore, we invite you to submit a revised version of the manuscript that addresses the points raised during the review process.

We look forward to receiving your revised manuscript.

Kind regards,

Bella Stevanny

Academic Editor

PLOS One

Journal Requirements:

Reviewers' comments:

Reviewer's Responses to Questions

**Comments to the Author**

1. If the authors have adequately addressed your comments raised in a previous round of review and you feel that this manuscript is now acceptable for publication, you may indicate that here to bypass the “Comments to the Author” section, enter your conflict of interest statement in the “Confidential to Editor” section, and submit your "Accept" recommendation.

Reviewer #1: All comments have been addressed

Reviewer #2: All comments have been addressed

Reviewer #3: All comments have been addressed

2. Is the manuscript technically sound, and do the data support the conclusions?

Reviewer #1: Partly

Reviewer #2: (No Response)

Reviewer #3: Yes

3. Has the statistical analysis been performed appropriately and rigorously? 

Reviewer #1: No

Reviewer #2: (No Response)

Reviewer #3: Yes

4. Have the authors made all data underlying the findings in their manuscript fully available?

Reviewer #1: No

Reviewer #2: (No Response)

Reviewer #3: Yes

5. Is the manuscript presented in an intelligible fashion and written in standard English?

Reviewer #1: No

Reviewer #2: (No Response)

Reviewer #3: Yes

6. Review Comments to the Author

Reviewer #1: Abstract:

Mistake/Weakness: The hazard ratio in the Abstract (HR 1.42) differs slightly from the one reported in the main text (Table 2, HR 1.421). While minor, consistency is key. The rounded value should be the same (1.42).

Weakness: The conclusion states, "Hysterectomy does not appear to significantly alter ovarian cancer risk..." This phrasing, while technically true based on the p-value, is misleading. A more accurate summary would be, "This study found no statistically significant association, but the wide confidence intervals preclude a definitive conclusion on whether hysterectomy increases, decreases, or has no effect on risk." The authors do mention "inconclusive findings" in the next sentence, but the primary conclusion is softly stated as "no effect."

2. Introduction

Weakness: The biological rationale is somewhat one-sided. The introduction focuses heavily on the hypothesis that hysterectomy might reduce ovarian cancer risk (via reduced ovarian function/ovulation). However, it largely ignores plausible mechanisms for a null or even increased risk (e.g., compensatory estrogen-only MHT use, as they later mention in the discussion, or the fact that simple hysterectomy leaves the primary site of origin for high-grade serous cancer—the fallopian tubes—in place). This sets up a expectation of a protective effect that the study then fails to find.

3. Methods

Study Design & Database: Well-described and appropriate.

Participant Selection:

Weakness: The "non-hysterectomy" group is defined as women who underwent a health checkup. This introduces a "healthy user" bias. Women who attend regular checkups may be systematically different from those who do not (e.g., more health-conscious, higher socioeconomic status), which could independently influence their ovarian cancer risk.

Outcomes:

Weakness: Defining ovarian cancer as "three or more visits... with a diagnosis code" is a good attempt to increase specificity and reduce false positives. However, the validity of this algorithm is not tested or referenced. Without linkage to a cancer registry to confirm the diagnosis, some misclassification is possible.

Statistics:

Major Weakness - Power Calculation: There is no mention of a pre-study power or sample size calculation. This is a critical omission. Given the low incidence of ovarian cancer, the authors should have estimated the number of events needed to detect a clinically meaningful hazard ratio (e.g., HR=0.7 or 1.3) before conducting the study. The fact that they ended up with only 48 total events is the fundamental limitation of this paper, and the lack of a power calculation makes it seem like this was an unforeseen problem rather than a known limitation of the study design from the outset.

Weakness (Addressed but noteworthy): The use of SMD < 0.2 as a balance threshold is standard, but the reviewer correctly pointed out that an SMD of 0.174 for age is not ideal. The authors' defense is statistically correct but clinically, a residual imbalance in the most important confounder (age) is not trivial.

4. Results

Weakness: The presentation of the main result is statistically correct but epidemiologically problematic. Stating a Hazard Ratio of 1.421 (suggesting a 42% increased risk) with a confidence interval of 0.79–2.56 is not evidence of "no effect." It is evidence of extreme uncertainty. The data are compatible with everything from a 21% risk reduction to a 156% risk increase. The Kaplan-Meier curve visually supports this high degree of uncertainty, with the lines virtually overlapping.

Major Weakness - Subgroup Analyses: The subgroup analyses are severely underpowered and should not have been performed, or their results should be dismissed. For example, the subgroup "Hysterectomy with adnexal surgery" has only 3 events, yielding a nonsensical HR of 1.5 with a CI of 0.251–8.977. Presenting these results in a table gives them a false sense of validity. They are uninterpretable.

5. Discussion

Strengths: The discussion does a good job of comparing findings to previous meta-analyses and proposing plausible explanations for the null result (shorter follow-up, MHT use).

Weaknesses:

Over-interpretation: The opening sentence, "this study found that hysterectomy is not significantly associated with ovarian cancer," is an overstatement. A more accurate phrasing would be "this study did not find a statistically significant association."

Ignoring the Point Estimate: The discussion focuses on the lack of significance and the wide CIs but does not adequately address the point estimate of HR=1.42. If this is a true effect, it would be clinically important. The possibility of an increased risk should be discussed more thoroughly, not just the failure to find a decreased risk.

Limitations Section: The limitation regarding the "limited number of ovarian cancer events" is correctly identified as the primary weakness. However, it should be stated more forcefully: "The study was severely underpowered to detect anything but a very large effect size, which fundamentally limits the conclusiveness of our findings."

Reviewer #2: (No Response)

Reviewer #3: This study addresses an important gap in gynecologic oncology literature regarding the long-term association between hysterectomy and ovarian cancer risk, specifically within the Asian population (South Korea). Utilizing the NHIS database provides a substantial sample size and minimizes recall bias. However, there are significant concerns regarding the definition of exposure groups, statistical power, and the interpretation of the "non-significant" findings which paradoxically lean towards an increased risk (HR > 1.0), contradictory to the protective mechanisms extensively discussed.

1. Clarification of Exposure Definitions and "Adnexal Surgery" (Crucial)

Observation: In the Abstract (Line 29-30), the objective mentions "simple hysterectomy (with or without bilateral salpingo-oophorectomy, BSO)". However, "Simple Hysterectomy" medically implies removal of the uterus only. Including BSO in the definition of "Simple Hysterectomy" is terminologically incorrect and confusing.

Critical Issue: In the Discussion (Lines 216-217), the author reports: "hysterectomy with adnexal surgery (HR 1.5, 95% CI 0.251–8.977)."

If "adnexal surgery" implies Bilateral Salpingo-Oophorectomy (BSO), it is biologically implausible for the risk of ovarian cancer to increase (HR 1.5), unless these are cases of Primary Peritoneal Carcinoma (PPC) or Ovarian Remnant Syndrome.

If the ovaries were removed, how is the incidence defined?

Recommendation: The authors must rigorously define "adnexal surgery." Does it mean unilateral oophorectomy? Salpingectomy only? Or BSO? If it is BSO, the authors must explain the histology of the cancers found (e.g., are they actually peritoneal cancers miscoded as C56?).

2. Divergence Between Statistical Trends and Discussion Logic

Observation: The primary result is an HR of 1.42 (increased risk, though non-significant). Yet, the Discussion (Lines 221-237) extensively details the "tubal hypothesis" and "blood flow disruption," both of which support a protective effect (decreased risk).

Critique: There is a logical disconnect. The authors spend significant text explaining why hysterectomy should reduce risk, while their data suggests a trend towards increased risk.

Recommendation: The discussion needs to be balanced. The authors should address why their point estimate (1.42) differs from the protective effects (HR < 1.0) seen in older meta-analyses. Could this be due to Indication Bias? For example, women undergoing hysterectomy for endometriosis or adenomyosis might have an inherently higher baseline risk of ovarian cancer, which the Propensity Score Matching (PSM) might not have fully accounted for if pathology data was lacking.

3. Statistical Power and Interpretation of "Inconclusive Findings"

Observation: The Abstract concludes findings are "inconclusive" due to wide Confidence Intervals (CI 0.79–2.56) and limited cases.

Critique: With 13,059 matched pairs, the study is relatively large, but the event rate is very low (incidence 18 vs 13 per 100,000). The extremely wide CI in the adnexal surgery subgroup (CI 0.251–8.977) renders that specific sub-analysis statistically useless.

Recommendation: Avoid over-interpreting the subgroup analyses. Acknowledging the "Type II Error" (false negative) probability is necessary. The authors should explicitly state in the Limitations that despite the large n, the study is underpowered for the specific outcome of ovarian cancer due to its rarity.

4. The "Temporal Shift" Argument

Observation: The Discussion (Lines 247-259) cites a meta-analysis suggesting a shift where post-2000 studies show a positive association (HR > 1).

Critique: This is the most interesting part of the literature review and aligns with the authors' own findings (HR 1.42).

Recommendation: Instead of focusing on the protective "tubal hypothesis," the authors should pivot the discussion to expand on this "temporal shift." Why are post-2000 studies showing increased risk? Is it better detection? Changes in surgical technique (e.g., sparing tubes in the past vs. opportunistic salpingectomy now)? Or is it that benign indications for hysterectomy (like endometriosis) are now better understood as risk factors for ovarian cancer?

Minor Comments

ICD-10 Code Specificity: The abstract mentions "C56.xx" (Line 35). Does this exclude Borderline Ovarian Tumors (BOT)? Inclusion of BOTs can inflate survival and incidence rates. Please clarify if the study is strictly invasive epithelial ovarian cancer.

Follow-up Duration: The median follow-up is 11.5 years. The discussion (Lines 283-285) mentions that protective effects might be seen after 20 years. The authors should explicitly recommend a longer follow-up study as a future direction.

Terminology: In Line 41, the term "adnexal surgery" is vague. Use precise terms like "Unilateral Salpingo-Oophorectomy" or "Bilateral Salpingectomy" throughout the manuscript.

Typo/Formatting: Line 215 states "HR 1.421" and Line 41 says "1.42". Please ensure consistent decimal places (2 decimal places is standard for HR reporting in abstracts).

7. PLOS authors have the option to publish the peer review history of their article (what does this mean?). If published, this will include your full peer review and any attached files.

Reviewer #1: No

Reviewer #2: **Yes:** Dr. Widad Akreyi

Reviewer #3: No

---

## [Author Response · Author response to Decision Letter 2]

14 Jan 2026

We thank for their careful reading, constructive feedback, and positive remarks about the significance and potential of our study. We respond to each point individually below, providing a robust and evidence-based defense for our methods and addressing all editorial and methodological comments.

we upload our review as an attachment because it exceeds 20,000 characters.

---

## [Decision Letter · Decision Letter 2]

12 Apr 2026

Effect of hysterectomy on the risk of ovarian cancer: A South Korean national cohort study

PONE-D-25-36133R2

Dear Dr. Yoon,

We’re pleased to inform you that your manuscript has been judged scientifically suitable for publication and will be formally accepted for publication once it meets all outstanding technical requirements.

Kind regards,

Mena Abdalla

Academic Editor

PLOS One

---

## [Editor Report · Acceptance letter]

PONE-D-25-36133R2

PLOS One

Dear Dr. Yoon,

I'm pleased to inform you that your manuscript has been deemed suitable for publication in PLOS One. Congratulations! Your manuscript is now being handed over to our production team.

Kind regards,

on behalf of

Dr. Mena Abdalla

Academic Editor

PLOS One